# Improvement of Arterial Stiffness One Month after Bariatric Surgery and Potential Mechanisms

**DOI:** 10.3390/jcm10040691

**Published:** 2021-02-10

**Authors:** Anna Oliveras, Isabel Galceran, Albert Goday, Susana Vázquez, Laia Sans, Marta Riera, David Benaiges, Julio Pascual

**Affiliations:** 1Nephrology Department, Hospital Universitari del Mar, 08003 Barcelona, Spain; igalceran@hospitaldelmar.cat (I.G.); svazquez@hospitaldelmar.cat (S.V.); lsans@hospitaldelmar.cat (L.S.); JPascualSantos@hospitaldelmar.cat (J.P.); 2IMIM (Hospital del Mar Medical Research Institute), 08003 Barcelona, Spain; agoday@hospitaldelmar.cat (A.G.); mriera1@imim.es (M.R.); dbenaiges@hospitaldelmar.cat (D.B.); 3Department of Experimental and Health Sciences, Area of Medicine, Universitat Pompeu Fabra, 08002 Barcelona, Spain; 4Red de Investigación Renal (REDINREN), Instituto Carlos III-FEDER, 28029 Madrid, Spain; 5Endocrinology Department, Hospital Universitari del Mar, 08003 Barcelona, Spain; 6Medicine Department, Universitat Autònoma de Barcelona, 08193 Barcelona, Spain

**Keywords:** bariatric surgery, arterial stiffness, renin-angiotensin axis

## Abstract

Arterial stiffness (AS) is an independent predictor of cardiovascular risk. We aimed to analyze changes (Δ) in AS 1-month post-bariatric surgery (BS) and search for possible pathophysiological mechanisms. Patients with severe obesity (43% hypertensives) were prospectively evaluated before and 1-month post-BS, with AS assessed by pulse-wave velocity (PWV), augmentation index (AIx@75) and pulse pressure (PP). Ambulatory 24 h blood pressure (BP), anthropometric data, renin-angiotensin-aldosterone system (RAAS) components and several adipokines and inflammatory markers were also analyzed. Overall reduction in body weight was mean (interquartile range (IQR)) = 11.0% (9.6–13.1). A decrease in PWV, AIx@75 and PP was observed 1-month post-BS (all, *p* < 0.01). There were also significant Δ in BP, RAAS components, adipokines and inflammatory biomarkers. Multiple linear regression adjusted models showed that Δaldosterone was an independent variable (B coeff.95%CI) for final PWV (B = −0.003, −0.005 to 0.000; *p* = 0.022). Angiotensin-converting enzyme (ACE)/ACE2 and ACE were independent variables for final AIx@75 (B = 0.036, 0.005 to 0.066; *p* = 0.024) and PP (B = 0.010, 0.003 to 0.017; *p* = 0.01), respectively. There was no correlation between ΔAS and anthropometric changes nor with Δ of adipokines or inflammatory markers except high-sensitivity C-reactive protein (hs-CRP). Patients with PWV below median decreased PWV (mean, 95%CI = −0.18, −0.25 to −0.10; *p* < 0.001) and both AIx@75 and PP at 1-month, but not those with PWV above median. In conclusion, there is an improvement in AS 1-month post-BS that correlates with ΔBP and Δrenin-angiotensin-aldosterone components. The benefit is reduced in those with higher PWV.

## 1. Introduction

The World Health Organization reported that more than 1.9 billion adults were overweight and, of these, over 650 million were obese [1]. Obesity is a well-established contributor to cardiac and all-cause mortality, independently of other associated cardiovascular risk factors [2,3]. Bariatric surgery (BS) consistently has shown to reduce cardiovascular morbidity and overall mortality [4,5], although the underlying mechanisms continue to be investigated.

Arterial stiffness (AS), considered as an independent cardiovascular risk factor [6], is a decrease in the ability of an artery to expand and contract in response to a given pressure change [7]. AS can be measured in many different ways [8]: pulse pressure (PP), pulse wave velocity (PWV), given that one of the fundamental principles of vascular pathophysiology is that pulse waves travel faster in stiffer arteries, and augmentation index (AIx), that expresses the degree of “augmentation” of central systolic blood pressure (SBP) as a consequence of systolic pressure waves travelling back to the heart and being received in late systole. Although these three indices are frequently used as AS markers, the most reliable seems to be PWV. It has been shown that PWV predicts mortality and cardiovascular outcomes [9], even independently of the Framingham Risk Score, showing better survival of individuals whose PWV responded to antihypertensive treatment independently of SBP reduction [10]. Moreover, high PWV is associated with increased cardiovascular disease risk regardless of hypertension status [11].

Excess body weight is associated with higher aortic stiffness in young and older adults [12]. Therefore, increased AS may be one of the mechanisms by which obesity increases cardiovascular risk independently of traditional risk factors. It is generally accepted that body weight decrease either by lifestyle intervention or by BS results in a reduction in AS. Some studies have reported a significant decrease In PWV or AIx at 3 months [13], 6 months [14,15], or beyond two years [16] after the intervention. There is no evidence or negative results regarding changes in PWV after weight loss in earlier stages [13,17].

The previously published BARIHTA (Hemodynamic Changes and Vascular Tone Control after Bariatric Surgery. Prognostic Value Regarding Hyper Tension and Target Organ Damage) study [18] analyses haemodynamic changes after BS. Here, changes in AS markers are analyzed, mainly as early as one month after BS. Additionally, we explore the role of different mechanisms potentially responsible for such changes.

## 2. Materials and Methods

### 2.1. Methods

#### 2.1.1. Study Design and Patients

The BARIHTA study is a prospective observational trial in a cohort of consecutively recruited patients with severe obesity scheduled to undergo BS (clinicaltrials.gov identifier: NCT03115502). Details about BARIHTA trial have been published elsewhere [18]. In brief, the BARIHTA study enrolled outpatients attending consults in the Hospital del Mar (Barcelona, Catalonia, Spain), because of severe obesity and looking for surgical treatment. All participants of both sexes aged 18–60 years with medical indication for treatment with BS and who agreed to undergo the surgical intervention, were invited to participate. Both normotensive and hypertensive patients were included. The exclusion criteria comprised the exclusion of the BS program for any reason or the refusal to give informed consent. The trial was approved by the local institutional Ethic Committee in accordance with the Declaration of Helsinki.

Here, we report additional analysis focused on the effect of BS on AS and its relationship with several renin-angiotensin-aldosterone system (RAAS) components, as well as with inflammatory markers and adipokines, according to pre-specified secondary endpoints.

Demographic and clinical data were recorded from all participants in the inclusion visit. Anthropometric characteristics, pharmacological treatment and 24 h blood pressure (BP) recordings, including data on PWV and AIx, and laboratory tests were obtained at baseline and 1, 3, 6 and 12 months after surgery. Hypertension was considered if patient received antihypertensive drugs and/or if the baseline 24 h-BP was ≥130/80 mmHg. Diabetes mellitus (DM) was diagnosed if the patient was under antidiabetic treatment or had ≥2 fasting plasma glucose determinations ≥7.0 mmol/L or if glycosylated haemoglobin A1c was >6.5%.

#### 2.1.2. Procedures

##### Mobil-O-Graph^®^ Device and Measurements

A Mobil-O-Graph^®^ NG-ambulatory blood pressure (NG-ABPM) by IEM, Stolberg, Germany device was used to measure brachial-BP and indirectly calculate aortic-BP and other arterial parameters through the oscillometric method (ARCSolver algorithm). Several studies have validated this device for estimating PWV and AIx [19,20]. Using suitable sized cuffs, the monitor was placed at 08:00–10:00 h A.M., and brachial artery waveforms were automatically recorded at 20-min intervals. Then, a generalized transfer function is applied to the averaged waveform to generate a corresponding aortic waveform. AIx was calculated as the ratio of the difference between the second systolic peak and the diastolic pressure and the difference between the first systolic peak and the diastolic pressure × 100. AIx was corrected for heart rate at 75 beats/min (AIx@75), as is standard [19,21,22]. The device also provided an indirect estimation of cardiac output.

All patients had recordings of good technical quality (≥70% valid readings). Otherwise, a new ambulatory-BP-monitoring (ABPM) was repeated within 1-week and used as the valid one.

##### Renin-Angiotensin-Aldosterone System (RAAS) Components

Plasma renin activity (PRA) and plasma aldosterone concentration, as well as angiotensin-converting enzyme (ACE) and angiotensin-converting enzyme-2 (ACE2) activities, were measured by validated laboratory methods [23]. Details on assay performance are reported in Appendix B.

##### Adipokines and Inflammatory Parameters

Leptin, adiponectin, and some cytokines and inflammatory markers, i.e., resistin, angiopoietin-2, MCP-1 and high-sensitivity C-reactive protein (hs-CRP) were also determined. See Appendix C.

##### Surgical Techniques

Either laparoscopic Roux-en-Y gastric bypass (LRYGB) or laparoscopic sleeve gastrectomy (LSG) were chosen for each patient based on clinical criteria and the consensus of the Bariatric Surgery Unit. Thus, LSG was preferred in younger patients, in those with BMI ranged 35–40 kg/m^2^, as a first-step treatment in cases with a body mass index (BMI) > 50 kg/m^2^ and when drug malabsorption was to be avoided [24]. The LRYGB technique involved a 150-cm antecolic Roux limb with 25-mm circular pouch–jejunostomy and exclusion of 50 cm of the proximal jejunum. In LSG, the longitudinal resection of the stomach from the angle of His to approximately 5 cm proximal to the pylorus was performed using a 36-French bougie inserted along the lesser curvature.

#### 2.1.3. Statistical Analyses

Descriptive data are presented as mean ± standard deviation (S.D.) for those normally distributed variables or summarized as median (interquartile range, IQR) in case of a non-normal distribution according to the Kolmogorov–Smirnov test. Categorical and dichotomous variables are presented as frequencies and percentages. Comparisons of variables between two periods were carried out by paired *t*-tests in continuous normally distributed data or by Wilcoxon-test in non-normally distributed continuous data. Multiple linear regression models were constructed for the resulting 1-month value of each AS parameter (dependent variable) adjusting for age, sex, variation (Δ) of body weight, Δ 24 h-systolic BP, Δ cardiac output, the baseline value of the correspondent AS marker and Δ of each assessed RAAS components (independent variables). Results are shown by the B coefficient and corresponding 95% confidence intervals (95%CI). Pearson’s or Spearman’s correlation coefficients, when appropriate, were obtained to measure the association between AS indexes and BP estimates, RAAS components, adipokines, inflammatory markers and glucose homeostasis parameters.

A sample size calculation was initially calculated and 61 subjects were assumed as needed to answer the primary outcome [18]. A post-hoc power calculation was performed regarding the paired *t*-test for the variable Δ 24 h-PWV (1-month after BS vs. baseline), the secondary endpoint analyzed here, having a sample size of 47 patients. Since we have a mean difference of −0.26 and a standard deviation of 0.40 (effect size = −0.65), for an α = 0.05, a power of 99.4% was found.

Statistical package SPSS for Windows version 25.0 (Cary, NC, USA) was used. A change was considered significant if the two-side alpha level was ≤0.05.

A quarter of the study population received treatment with one or more drugs that interfered with RAAS, being modified within the first month post-BS. For this reason, the main analyses, especially those which include BP and/or RAAS parameters, were performed separately in both the whole cohort and in the normotensive patients.

## 3. Results

Sixty-two patients completed the BARIHTA study. Complete data on AS were available for 47 subjects, and these comprise the cohort reported here (a flowchart is supplied in Appendix A). Main baseline clinical characteristics are described in Table 1.

There was a higher prevalence of female patients, and 43% of individuals were hypertensives. None of the patients died in the follow-up period. Of note, no patient in this study was treated with a sodium-glucose transport protein 2-inhibitor or a glucagon-like peptide-1 receptor agonist.

### 3.1. Changes in Anthropometric and Hemodynamic Parameters

As regards the primary outcome of the BARIHTA study, there was a 12-month decrease in both 24 h-central and 24 h-peripheral SBP (mean, 95% CI) of −4.4 mmHg (−8.3 to −0.5) and −4.0 mmHg (−7.8 to −0.2) respectively.

Changes (Δ) in AS were statistically significant 1-month post-BS (Table 2).

Although there was a trend towards a decrease in all AS markers from baseline to 3, 6 and 12 months [25], only the decrease of 24 h PP at 12 months was statistically significant: mean (95%CI) = −2.1 mmHg (−4.1 to −0.1), *p* = 0.042. Since our goal was to analyze changes in AS and their potential mechanisms, most of the analyses shown from now on are referred to the evaluation 1-month post-BS. Thus, 1-month changes in anthropometric and hemodynamic parameters and arterial stiffness estimates are shown in Table 2, both in the whole cohort and after excluding patients with any antihypertensive treatment, and for all these variables there was a statistically significant decrease (*p* < 0.01 for all).

Statistically significant decreases in body weight and waist circumference are also shown in Table 2. The overall reduction in body weight was mean (IQR) = 11.0% (9.6–13.1).

### 3.2. Changes in Glucose Metabolism, RAAS Components, Adipokines and Inflammatory Markers

There was a statistically significant decrease in all glucose metabolism parameters (*p* < 0.001) (Table 2).

Table 2 also describes the baseline and 1-month post-BS values of the RAAS components, showing statistically significant decreases in both ACE and ACE2 activities. However, none change was observed in either PRA or plasma aldosterone concentration values. Appendix A shows that decreases in PRA and plasma aldosterone concentration occur from 3-months on. As regards adipokines and inflammatory markers, there was a statistically significant decrease in leptin and a trend towards a decrease in hs-CRP, as well as increases in other adipokines (adiponectine, MCP-1 and angiopoietin-2).

When analyzing anthropometrics, arterial stiffness and hemodynamic changes according to the surgical technique (Appendix A), to having or not sleep-apnea or to sex [25], no between-groups remarkable differences were observed.

Multiple linear regression models were built for each of the statistically significant change in AS markers (Table 3).

In all tested models for final (1-month post-BS) 24 h PP and final PWV, Δ 24 h systolic BP and baseline values of each correspondent AS marker were statistically significant independent variables. Another independent variable for 24 h PP post-BS was Δ ACE. On the other hand, Δ aldosterone was an independent variable for the final PWV value. As regards final AIx@75, age and baseline AIx@75 value were statistically significant independent variables in all models. In addition, the ratio ACE/ACE2 was also an independent variable for the final AIx@75 value in this cohort. Neither Δ body weight nor Δ cardiac output influenced the final value of any AS marker. Equivalent results were found when the same models were tested including Δ waist circumference instead of Δ body weight [25]. 

In addition, Pearson’s or Spearman’s correlation coefficients, as appropriate, were obtained to measure the association between the observed changes.

### 3.3. Correlations

#### 3.3.1. Correlations of Changes in Arterial Stiffness (AS) with Changes in Anthropometric Parameters, Glucose Metabolism, Adipokines and Inflammatory Markers

At 1-month, there was no statistically significant correlation between changes in PWV, AIx@75 or PP with changes in body weight or waist circumference. When the correlations between the variation of each of these three AS markers with changes in fasting glucose, fasting insulin or the HOMA-IR index were explored, again no correlation was found.

In addition, none of the changes in any of the analyzed adipokines or inflammatory markers showed a statistically significant correlation with changes in PWV, AIx@75 or PP. except between variation of 24 h PP and variation of hs-CRP: Rho = 0.382; *p* = 0.041 [25].

#### 3.3.2. Correlations of Changes in AS with Changes in BP Estimates

Variation of 24 h-PWV (Figure 1A) correlated with Δ 24 h-SBP (Pearson’s coefficient = 0.384; *p* = 0.036. No statistically significant correlation was observed between ΔAIx@75 and any BP estimate.

#### 3.3.3. Correlations of Changes in AS with Changes in the RAAS Components

Variation of 24 h PWV correlated with Δ ACE/ACE2 ratio (Pearson’s coefficient = −0.488; *p* = 0.013). There was also a direct correlation between Δ AIx@75 and Δ ACE (Pearson’s coefficient = 0.435; *p* = 0.026) (Figure 1B). At 1-month there was no correlation between changes in PRA or aldosterone levels and changes in any AS marker. On the other hand, there was no correlation between changes in aldosterone levels and changes in ACE activity.

Finally, given the fact that some authors [12] reported significant improvement of AS after BS only in those with pathological preoperative PWV, we explored three quantile regression models for changes in AS markers. Therefore, we segmented the sample into two subsets above and below the median of baseline 24 h PWV, AIx@75 and 24 h-PP, respectively (Table 4).

Patients with higher baseline 24 h PWV only showed a statistically significant decrease in PWV and AIx@75 1-month post-BS, but not in the PP. In contrast, those with 24 h-PWV below the median showed a decrease in all AS markers at 1-month. Conversely, patients with lower AIx@75 and lower PP only showed an improvement in 24 h PWV at 1-month. Remarkably, for patients with AIx@75 or PP above the median, statistically significant decreases were observed in all AS markers at 1-month.

## 4. Discussion

Arterial stiffness, assessed by three different methods, i.e., PWV, AIx@75 and PP, improved significantly as early as 1-month after BS in this cohort of patients with severe obesity. These changes were confirmed in the subset of patients strictly normotensives, as confirmed by the 24 h -ABPM, or who did not experience changes in their antihypertensive treatment regimen. In fact, it has been suggested that AS may precede elevations in systolic BP and incident hypertension in obese individuals [26]. Moreover, although there are several reports observing an improvement in AS after losing weight, these changes have demonstrated to be significant only from 3 months after BS, as summarized by Petersen [7] et al. In this systematic review and meta-analysis, where all studies had a follow-up time of more than 1-month, it is shown that although some AS measures improved 3 months after weight loss, these changes were not observed according to AIx or PP, while PWV was not evaluated. Even in a very recent study on this issue [12], no significant decrease in AS was observed 1-month post-BS except in patients with preoperative pathologic PWV. And another group reported very recently significant changes in PWV at 8 months after BS [27]. Along these lines, the second important point is that we have demonstrated that main changes in the different markers of AS occur in patients with lowest PWV, suggesting that perhaps PWV is a stronger marker of organ damage than AIx@75 or PP, and thus less susceptible to modification even after losing weight. On the contrary, for patients with AIx@75 or PP above the median, statistically significant decreases were observed in all AS markers. The set of these findings suggests that PWV is a more powerful marker of AS and less likely to change, whereas elevated AS based on AIx@75 or PP is more likely to improve after BS, and we believe that it adds knowledge to what is reported until now on relationships between obesity, AS and changes after BS. Thus, although PWV is the gold standard for AS measurement [21], perhaps AIx and PP should receive more attention as modifiable therapeutic targets, at least in the obese population. It is known that PWV closely relates to arterial wall stiffness whereas AIx is related to both arterial wall stiffness and wave reflection, which is dependent on peripheral resistance and affected by heart rate variation. This finding is in accordance with that reported by Rossi et al. [28] in primary aldosteronism, where it was suggested that vascular damage may be partially reversible and that both forward and backward pulse wave amplitudes might be more accurate than PWV to detect subtle changes of function in large arteries. It is probable that this difference between the two AS markers would justify the fact that AIx is a more modifiable parameter than PWV, given the changes in heart rate and peripheral resistances are associated with weight loss. We must also highlight the novelty of assessing AS parameters by an oscillometric device providing 24 h measurements. In fact, the vast majority of reports regarding changes in AS after BS use “office” methods, such as applanation tonometry. Here we have shown changes in 24 h ambulatory measurements, which add value to the findings, although confirmation in further studies would be desirable. Thirdly, there were significant changes in anthropometric parameters, glucose metabolism components and adipokines and inflammatory markers. However, none of them showed a statistically significant role in the observed AS improvement. Otherwise, and as expected, BP determined final PWV, while the main independent variable for final AIx@75 was age. Regarding the potential mechanisms responsible for the reduction of AS after weight loss, some authors [13] have found a correlation between weight loss and reduction of PWV independently of changes in established hemodynamic and cardiometabolic risk factors. Other groups [15], but not all [12], suggest that this correlation is mediated by the decrease in BP. Aside from conflicting reports on the role of BP in AS, elevated cardiac volume and output in obese individuals have also been noted as possible mediators of AS, more important than elevated BP [29]. Given the non-significant changes in PWV at 3, 6, and 12 months, also reported by Cooper et al. [14], we must keep in mind that AS is influenced by both functional and structural factors. We hypothesize that hemodynamic changes occur primarily in the first few weeks after BS, from which they are likely to stabilize, while surely the deepest structural changes remain. This could justify the lack of a permanent decrease of the PWV. In relation to carbohydrate metabolism, there are contrasting results regarding changes in glucose metabolism and insulin sensitivity parameters and AS modification: while some studies show a relationship with AIx [14], others suggest that there is no correlation [16].

What is relevant is the finding that changes in several RAAS components were also independent variables for the final values of each AS markers, after adjusting for confounding factors, relationships that were confirmed in correlation analyses. Some studies have previously suggested that the RAAS is an important determinant of AS, in addition to BP and other factors. Multiple mechanisms are responsible for the RAAS activation in obesity, including adipose tissue-derived RAAS components that might be involved in regulation of BP and AS through local production of angiotensin (Ang) II and aldosterone, conversion of adipocyte-derived angiotensinogen by systemic renin and ACE-activity, or forming Ang via alternative routes due to the presence of cathepsins and chymase in human adipose tissue [27,30]. The main changes in this BARIHTA study were observed in ACE and ACE2, but no significant change was found in PRA or plasma aldosterone concentration 1-month post-BS, nor were there any correlations between their changes and improvement on any AS marker. Surprisingly, aldosterone concentration did not change, but its variation was shown to determine PWV changes at 1 month, although, as mentioned, the decrease in aldosterone levels began at 3 months. Perhaps this relationship is statistically significant due to its inclusion in a model with other variables, but from our point of view, what is really important is that the RAAS components have an overall impact on AS, as can be deduced from the three models. There is a predominant role of adipose-tissue-derived RAAS components in the development of AS in obese individuals and its consequent improvement after BS. All components of RAAS, except renin, are known to be found in aortic and mesenteric perivascular adipose tissue, including angiotensinogen, ACE, ACE2, chymase, Ang I, Ang II, AT1, and AT2 receptors [31]. Other authors [32] have also suggested that the generation of some RAAS components through a non-renin-dependent pathway is likely. Taking together, this may explain why changes in AS are related to certain components of RAAS, but not to PRA or aldosterone, at least one month after BS. Moreover, it justifies why there is no correlation between changes at 1-month in ACE activity and changes in aldosterone concentration since, as described, the decrease in the latter occurred from 3 months. Another point to address is the finding that ACE2 activity is elevated at baseline and decreases after BS, considering that its metabolite Ang (1–7) exerts inhibitory effects on inflammation and on vascular and cell growth mechanisms. Recent studies have shown that increased activation of ACE2/Ang-(1–7)/MasR axis can revert and prevent local and systemic dysfunctions improving lipid profile and insulin resistance by modulating insulin actions, and reducing inflammation [32,33]. Therefore, the increased ACE2 would counterbalance the adverse effects of raised Ang II in obesity by increasing levels of the vasodilator Ang-(1–7), as has been shown in other pathological conditions, although it is only speculation. In addition, the increase in the ACE/ACE2 ratio suggests that the decrease in ACE2—probably overexpressed before surgery—is greater than the decrease in ACE, perhaps due to the faster normalization of a compensatory mechanism than that of a pathological one.

Our study has some limitations and several strengths. First, there is some debate about the reliability or not of the Mobile-O-Graph device for measuring AS in the general population and in patients with obesity or with Marfan’s syndrome [34]. Recently, a couple of studies have concluded that the oscillometric PWV of Mobil-O-Graph is explained almost entirely by age and SBP compared to carotid-femoral PWV [35] or to the invasive aortic PWV measurement [36], since their relationship is explained by shared associations with age and SBP. However, it has been established that estimated PWV can be used to improve risk prediction in addition to traditional risk classification in conditions under which measuring carotid-femoral PWV is not feasible [8]. Anyway, we have used this same method at different time points for each individual, which is why we truly believe that the modification over time of this parameter has value. On the other hand, due to an otherwise common underrepresentation of male patients in the analyzed cohort, it is not possible to explore the influence of gender on the relationships between RAAS components and AS in this setting. Anyway, we discarded any between-sex differences in 1-month changes in anthropometric and AS measurements and, on the other hand, sex was included in the regression analyses. Otherwise, although we have not explored other possible mechanisms for AS, such as the overproduction of reactive oxygen species or the role of the sympathetic nervous system, we have analyzed the most important mechanisms of its overactivation in obesity, i.e., hyperinsulinemia, hyperleptinemia, RAAS activation and the presence of obstructive sleep-apnea [29]. Finally, it is said that other central obesity indices, such as the waist-to-hip ratio, may be a superior predictor of obesity-related cardiometabolic risk than BMI. However, using the waist circumference we obtained results similar to those obtained with body weight. There are several relevant strengths. Most of the results reported here refer to patients with confirmed normal 24 h BP, except for a small proportion of never-treated hypertensives or patients who did not change antihypertensive treatment throughout the study. This is of high relevance because it emphasizes that in patients with severe obesity there are subtle structural alterations, even below the cutoff values accepted as normality, which improve after BS, indicating a possible higher cardiovascular risk for to this otherwise normotensive population. Moreover, we want to highlight the relative youth of this cohort, which makes the structural changes in the arteries even more relevant. Finally, the inclusion of ACE and ACE2 in this study may contribute to deepen the exploration of pathophysiological mechanisms on the topic we are dealing with, although our study does not allow to establish causal relationships. These data support the need for broader studies questioning the link of AS with RAAS to determine how sustained weight loss reduces cardiovascular morbidity and whether treatment with RAAS inhibitors could have an equivalent benefit.

## 5. Conclusions

Severely obese patients, including normotensives, have some degree of AS that improves one-month post-BS. Patients with the highest baseline PWV are less likely to improve, but improvement of AS in those with higher baseline AIx@75 maintains over time. AS changes are probably related to modifications in the RAAS, specifically to ACE and ACE2 activities, although this possibility deserves further investigation.

## Figures and Tables

**Figure 1 jcm-10-00691-f001:**
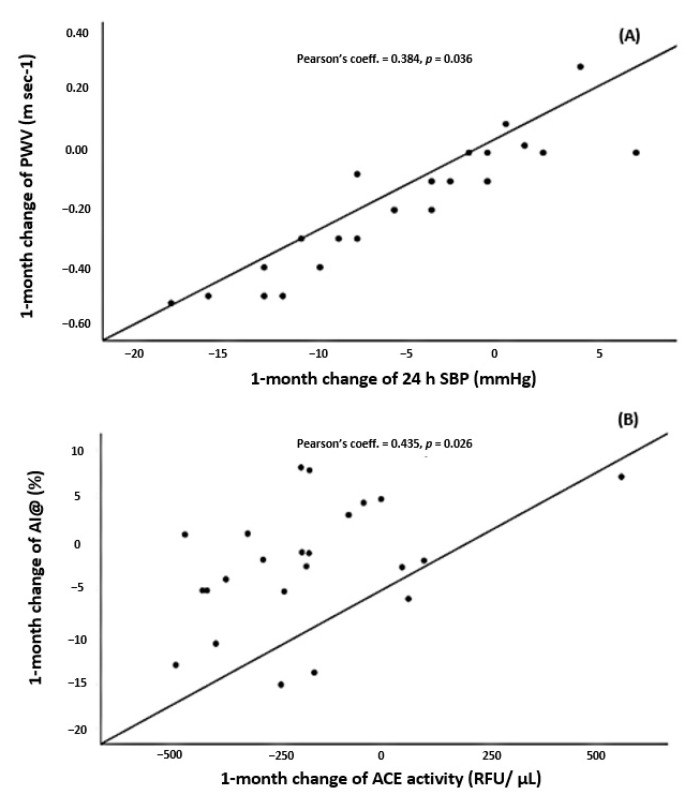
(**A**) Scatter plot for the correlation between change of PWV and change of 24 h SBP one-month after bariatric surgery. (**B**) Scatter plot for the correlation between change of AIx@75 and change of ACE one month after bariatric surgery. ACE = angiotensin converting enzyme; AIx@75 = augmentation index at 75 beats/minute; PWV = pulse-wave velocity; SBP = systolic blood pressure.

**Table 1 jcm-10-00691-t001:** Baseline clinical characteristics.

Age, year (mean ± SD)	42.7 ± 9.4
Sex, women, *n* (%)	34 (72.3)
Body weight, kg (mean ± SD)	118.1 ± 19.5
Waist circumference, cm (mean ± SD)	131.6 ± 10.7
Body mas index, kg/m^2^ (mean ± SD)	42.2 ± 5.4
Race, Caucasian, *n* (%)	45 (95.7)
Current smokers, *n* (%)	10 (21.3)
Surgical procedure, *n* (%):	
Sleeve gastrectomy	20 (42.6)
Roux-en-Y gastric bypass	27 (57.4)
Hypertension, *n* (%)	20 * (42.6)
T2-Diabetes Mellitus, *n* (%)	4 (8.5)
Dyslipidemia, *n* (%)	12 (25.5)
Previous major vascular event **, *n* (%)	3 (6.4)
Sleep apnea syndrome, *n* (%)	11 (23.4)
CPAP, *n* (%)	9 (81.8)

T2 = type 2; CPAP = continuous positive airway pressure. * Three of them, never-treated hypertensives. ** Coronary artery disease, heart failure, ictus or peripheral vascular disease.

**Table 2 jcm-10-00691-t002:** Changes 1-month after bariatric surgery.

	All Patients (*n* = 47)			Patients without Antihypertensive Treatment at Baseline ** (*n* = 30)		
Parameter	Baseline Mean ± SD	1-Month Mean ± SD	*p*	Baseline Mean ± SD	1-Month Mean ± SD	*p*
**Anthropometric parameters**						
Body weight, kg	118.1 ± 19.5	104.5 ± 17.5	<0.001	116.1 ± 17.7	102.5 ± 15.2	<0.001
Waist circumference, cm	132.0 ± 12.0	122.4 ± 10.2	<0.001	130.2 ± 10.6	120.2 ± 9.3	<0.001
**Arterial stiffness**						
24 h-PP, mmHg	46.4 ± 7.5	44.2 ± 7.1	0.001	45.8 ± 6.9	43.1 ± 6.6	0.010
24 h-PWV, m/s	6.64 ± 1.03	6.24 ± 0.97	<0.001	6.2 ± 1.0	6.11 ± 0.99	0.001
AIx@75, %	26.4 ± 7.5	22.7 ± 7.1	<0.001	24.8 ± 5.9	22.2 ± 6.0	0.028
**Blood pressure, heart rate and cardiac output**						
24 h-SBP, mmHg	120.0 ± 11.7	114.3 ± 9.9	<0.001	118.6 ± 10.7	113.5 ± 9.7	<0.001
24 h-DBP, mmHg	73.7 ± 9.0	70.2 ± 6.9	<0.001	72.1 ± 7.7	69.4 ± 6.8	<0.001
24 h-HR, bpm	73.2 ± 10.1	66.7 ± 8.8	<0.001	75.2 ± 9.2	68.7 ± 7.6	<0.001
Cardiac output	4.6 ± 0.6	4.5 ± 0.5	<0.001	4.7 ± 0.5	4.5 ± 0.4	0.004
**Glucose metabolism parameters**						
Fasting glucose, mg/dL	99.6 ± 19.1	86.5 ± 10.1	<0.001	95.9 ± 12.9	85.7 ± 9.2	<0.001
Glycosylated hemoglobin, %	5.7 ± 0.8	5.3 ± 0.6	<0.001	5.7 ± 1.0	5.3 ± 0.6	0.001
Fasting insulin *, mcU/mL	12.0 [8.3; 17.3]	6.8 [3.9; 9.4]	<0.001	11.7 [7.1; 17.3]	6.3 [2.9; 9.2]	<0.001
Insulin resistance (HOMA-IR) *	54.5 [37.2; 87.1]	23.8 [13.8; 39.3]	<0.001	50.4 [28.3; 82.2]	23.3 [10.3; 39.3]	<0.001
**RAAS components**						
PRA *, ng/mL/h	0.8 [0.3; 1.3]	0.8 [0.5; 1.2]	0.726	0.8 [0.4; 1.2]	0.9 [0.5; 1.5]	0.411
Aldosterone *, ng/dL	87.8 [56.8; 134.5]	82.0 [61.4; 139.5]	0.747	76.7 [59.3; 108.3]	86.0 [65.8; 128.5]	0.210
ACE activity, RFU/µL	1320.2 ± 385.8	1099.0 ± 293.7	<0.001	1307.9 ± 337.4	1126.6 ± 258.0	<0.001
ACE2 activity *, RFU/µL/h	7.9 [5.8; 10.8]	6.0 [4.7; 7.8]	<0.001	7.6 [5.5; 9.4]	5.8 [4.2; 7.8]	0.001
ACE act./ACE2 act.	170.0 ± 82.0	194.1 ± 108.7	0.009	183.0 ± 93.3	207.4 ± 118.1	0.072
**Adipokines & Inflammatory Markers**						
Leptin *, ng/mL	56.5 [28.8; 73.7]	21.4 [12.1; 37.3]	<0.001	45.1 [24.3; 67.1]	15.9 [9.5; 31.5]	<0.001
Adiponectine *, μg/mL	19.0 [12.5; 33.3]	23.6 [12.9; 40.5]	0.050	22.5 [16.0; 35.5]	27.1 [16.8; 45.6]	0.043
Resistin, ng/mL	36.8 ± 13.3	37.3 ± 14.9	0.822	35.9 ± 11.2	39.4 ± 12.7	0.106
MCP-1, pg/mL	544.9 ± 197.5	601.5 ± 249.4	0.041	530.0 ± 199.5	632.6 ± 274.7	0.014
Angiopoietin2 *, pg/mL	2676.2 [1815.4; 4067.0]	4039.6 [2076.8; 5380.9]	0.042	2721.4 [1815.4; 4655.1]	4366.8 [2011.9; 5293.3]	0.046
hs-CRP *, mg/dL	0.77 [0.43; 1.41]	0.45 [0.24; 0.74]	0.001	0.57 [0.32; 0.93]	0.56 [0.25; 0.74]	0.249

***** Data shown as median [interquartile range]. ** normotensive patients (*n* = 27) plus mild hypertensive patients without antihypertensive treatment (*n* = 3). ACE = angiotensin converting enzyme; ACE2 = angiotensin converting enzyme 2; AIx@75 = augmentation index at 75 beats/minute; DBP = diastolic blood pressure; HOMA = homeostasis model assessment; HR = heart rate; hs-CRP = C-reactive protein; MCP-1 = monocyte chemoattractant protein-1; PP = pulse pressure; PRA = plasma renin activity; PWV = pulse-wave velocity; RAAS = renin-angiotensin aldosterone system; RFU = relative fluorescence units; SBP = systolic blood pressure; ACE = angiotensin converting enzyme; ACE2 = angiotensin converting enzyme 2; AIx@75 = augmentation index at 75 beats/min.

**Table 3 jcm-10-00691-t003:** Role of several factors, including RAAS components, on changes in arterial stiffness markers in patients without antihypertensive treatment.

	Dep. Variable	B Coefficient	95% (CI)	*p*
Model 1	**1-month 24 h PP**			
	Age			
	Sex			
	Δ 24 h-SBP	0.481	0.272–0.689	0.001
	Δ body weight			
	Δ cardiac output			
	Baseline 24 h-PP	0.996	0.771–1.220	< 0.001
	Δ ACE			
Model 2	**1-month PWV**			
	Age			
	Sex			
	Δ 24 h-SBP	0.032	0.008–0.056	0.014
	Δ body weight			
	Δ cardiac output			
	Baseline PWV	0.651	0.416–0.886	< 0.001
	Δ aldosterone	−0.003	−0.005–0.000	0.022
Model 3	**1-month AI@75**			
	Age			
	Sex			
	Δ 24 h-SBP			
	Δ body weight			
	Δ cardiac output			
	Baseline AI@75	0.326	0.013–0.639	0.043
	Δ ACE/ACE2	0.036	0.005–0.006	0.024

Δ = change; ACE = angiotensin converting enzyme; ACE2 = angiotensin converting enzyme 2; AIx@75 = augmentation index at 75 beats/minute; PP = pulse pressure; PWV = pulse-wave velocity; SBP = systolic blood pressure. Adjusted squared R = 0.863 (Model 1), 0.900 (Model 2) and 0.689 (Model 3).

**Table 4 jcm-10-00691-t004:** Changes in AS markers according to the segmented population into two subsets above and below the median of baseline PWV, baseline AIx@75 and baseline 24 h PP, respectively.

	Below Median Baseline 24 h-PWV (*n* = 21)		Above Median Baseline 24 h-PWV (*n* = 15)		Below Median Baseline 24 h AIx@75 (*n* = 18)		Above Median Baseline 24 h-AIx@75 (*n* = 18)		Below Median Baseline 24 h-PP (*n* = 18)		Above Median Baseline 24 h-PP (*n* = 18)	
	Mean (95%CI)	*p*	Mean (95%CI)	*p*	Mean (95%CI)	*p*	Mean (95%CI)	*p*	Mean (95%CI)	*p*	Mean (95%CI)	*p*
**Δ 24 h-PWV**												
1 month	−0.18 (−0.25 to −0.10)	**<0.001**	−0.31 (−0.64 to 0.02)	0.060	−0.18 (−0.28 to −0.08)	**0.002**	−0.29 (−0.55 to −0.03)	**0.034**	−0.16 (−0.28 to −0.04)	0.013	−0.31 (−0.56 to −0.05)	**0.020**
**Δ 24 h-AIx** **@75**												
1 month	−3.8 (−6.7 to −0.9)	**0.014**	−1.6 (−4.6 to 1.3)	0.261	−0.0 (−2.8 to 2.7)	0.990	−5.7 (−8.3 to −3.2)	**<0.001**	−1.9 (−4.3 to 0.4)	0.106	−4.3 (−6.7 to −1.9)	**0.001**
**Δ 24 h-PP**												
1 month	−3.4 (−5.5 to −1.3)	**0.003**	−2.3 (−5.1 to 0.4)	0.563	−2.0 (−5.0 to 1.0)	0.177	−2.7 (−4.8 to −0.6)	**0.015**	−2.1 (−5.1 to 0.8)	0.149	−3.6 (−6.7 to −0.6)	**0.023**

Δ = change; AIx@75 = augmentation index at 75 beats/minute; PP = pulse pressure; PWV = pulse-wave velocity. Significant results are highlighted in bold format.

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
