# Peer review of "Improvement of Arterial Stiffness One Month after Bariatric Surgery and Potential Mechanisms"

_jcm, 2021, doi:10.3390/jcm10040691_

Round 1

Reviewer 1 Report

The Authors studied arterial stiffness after bariatric surgery. The results are really interesting. The paper can be accepted.

Author Response

Dear reviewer,

Thank you very much for enjoying our work!

Reviewer 2 Report

In their paper the authors studied the change in pulse wave velocity (PWV), augmentation index (AI) and pulse pressure (PP) before and 1 month after bariatric surgery in 47 patients, using a 24hour device (Mobil-O-Graph). They found a decrease in PWV, AI and PP, as well as in BP, adipokines and inflammatory biomarkers. After adjustment for several confounders they concluded that a change in aldosterone was an independent predictor of the final PWV, while ACE/ACE2 and ACE of the final AI and PP. The reduction in PWV was less in those with higher baseline PWV.

This is an interesting study; however some issues need to be addressed.

  • In introduction, the meta-analysis by Vlachopoulos et al regarding the predictive value of PWV on events and mortality should be added: DOI: 10.1016/j.jacc.2009.10.061.
  • In methods section it would be useful to add the validation studies of mobilograph for PWV, AI etc measurements (DOI: 10.1016/j.ijcard.2013.08.079, DOI: 10.1097/MBP.0b013e3283614168).
  • The authors should highlight how this study differs from other studies on the same topic (eg use of 24hour measurements with an oscillometric device) and what it adds to the existing literature regarding changes in arterial stiffness after bariatric surgery.
  • The change In PWV in one month (a change of 0.4 m/s) is statistically significant but clinically not so significant Also, as the authors state, no significant difference was found in 3, 6 and 12 months. How do the authors explain this finding?
  • In table 3 the authors could also add the change in insulin in multivariate models. HR? Also use change in BMI instead of change in weight?
  • Since the reduction of arterial stiffness indices at 3, 6 and 12 months were not statistically significant, the authors should state that also in the discussion and suggest possible explanations for this. 

Reviewer 3 Report

This study is interesting.  I  have following comments or suggestions, which maybe helpful to understanding of the mechanisms of improvement of arterial stiffness after bariatric surgery.

  1. Define and check the data ACE (RFU/μL),  ACE2 (RFU/µL/hr), and ACEact./ACE2act in table 2.  It is very hard to understand why after bariatric surgery, the ratio of ACEact/ACE2act was increased, which was related to the reduction of arterial stiffening. Please discuss.
  2. It is interesting know whether a change in ACE activity is associated a change in aldosterone concentration.
  3. Aldosterone was a key molecule to determine the stiffness of  arterial wall (Table 3), but there was no difference between treatment and baseline in Table 2. Please explain.
  4. Address the issue " the benefit of treatment is reduced in subjects with higher PWV in the section of DISCUSS.
  5. Spell out all abbreviations in the abstract such as ACE etc. 
  6. It is confusing about the number of never-treated hypertension listed in table 2 vs. table 1. Please double check.
  7. Clarify "previous major vascular event" in table 1.

Round 2

Reviewer 2 Report

The manuscript was substantially improved.

Reviewer 3 Report

I have no further comments.